# Spatiotemporal spread pattern of the COVID-19 cases in China

**Yongjiu Feng**  [1]*, **Qingmei Li**[1,2], **Xiaohua Tong**[1]*, **Rong Wang**[1,3], **Shuting Zhai**[1,3], **Chen Gao**[1,3], **Zhenkun Lei**[1,3], **Shurui Chen**[1,3], **Yilun Zhou**[1], **Jiafeng Wang**[1], **Xiongfeng Yan**[1], **Huan Xie**[1], **Peng Chen**[1], **Shijie Liu**[1], **Xiong Xv**[1], **Sicong Liu**[1], **Yanmin Jin**[1], **Chao Wang**[1], **Zhonghua Hong**[3], **Kuifeng Luan**[3], **Chao Wei**[1], **Jinfu Xu**[4], **Hua Jiang**[5], **Changjiang Xiao**[1], **Yiyou Guo**[1]

**1** College of Surveying and Geo-Informatics, Tongji University, Shanghai, China, **2** College of Engineering, Peking University, Beijing, China, **3** College of Marine Sciences, Shanghai Ocean University, Shanghai, China, **4** Shanghai Pulmonary Hospital, Tongji University, Shanghai, China, **5** Shanghai East Hospital, Tongji University, Shanghai, China

* yjfeng@tongji.edu.cn (YF); xhtong@tongji.edu.cn (XT)

## Abstract

The COVID-19 pandemic is currently spreading widely around the world, causing huge threats to public safety and global society. This study analyzes the spatiotemporal pattern of the COVID-19 pandemic in China, reveals China's epicenters of the pandemic through spatial clustering, and delineates the substantial effect of distance to Wuhan on the pandemic spread. The results show that the daily new COVID-19 cases mostly occurred in and around Wuhan before March 6, and then moved to the Grand Bay Area (Shenzhen, Hong Kong and Macau). The total COVID-19 cases in China were mainly distributed in the east of the Huhuanyong Line, where the epicenters accounted for more than 60% of the country's total in/on 24 January and 7 February, half in/on 31 January, and more than 70% from 14 February. The total cases finally stabilized at approximately 84,000, and the inflection point for Wuhan was on 14 February, one week later than those of Hubei (outside Wuhan) and China (outside Hubei). The generalized additive model-based analysis shows that population density and distance to provincial cities were significantly associated with the total number of the cases, while distances to prefecture cities and intercity traffic stations, and population inflow from Wuhan after 24 January, had no strong relationships with the total number of cases. The results and findings should provide valuable insights for understanding the changes in the COVID-19 transmission as well as implications for controlling the global COVID-19 pandemic spread.

## Introduction

At present, many countries have reported a high number of COVID-19 cases and the pandemic is raging around the world. As of August 20, the world has accumulated more than 20 million of the COVID-19 cases, threatening people's health, economic development and social stability. The origin and birthplace of the COVID-19 pandemic are still being explored by biologists around the world [1–3]. In the era of globalization, China has not been immune from

## PLOS ONE

**Data Availability Statement:** All COVID-19 datasets are available from the National Health Comission of China (http://www.nhc.gov.cn).

**Funding:** This work was financially funded by Tongji University's key program "Spatial big data-

based trajectory tracking and spread warning for better prevention and control of the COVID-19 epidemic " (Grant No. 22120200004) and the National Natural Science Foundation of China (Grant No. 42071371). The funder had no role in study design, data collection and analysis, decision to publish, or preparation of the manuscript.

**Competing interests:** The authors have declared that no competing interests exist.

the pandemic. In early January 2020, COVID-19 in China broke out in Wuhan, the capital of Hubei Province, and quickly spread throughout China during the Chinese New Year [4]. China had confirmed approximately 84,000 cases and 4,642 deaths as of April 20 (GMT 24:00 +8), according to the National Health Commission, China [5]. China has now successfully controlled the spread of COVID-19, but still needs to strictly prevent and treat the overseas imported cases.

With the rapid spread of COVID-19 pandemic in the world, it is of great significance to analyze its spatial evolution in China, especially the transmission pattern and spread trend, improving our understanding of the spatiotemporal mechanisms. Some have applied mathematical models to simulate and project the spread of COVID-19 [6–8]. Li et al. conducted a preliminary evaluation of the epidemiological characteristics of COVID-19 [9], indicating that human-to-human transmission caused by close contacts began to emerge in mid-December and gradually spread over the following month. Subsequently, stochastic transmission models with the COVID-19 cases were established to quantify the effectiveness of contact tracking, cases separation and travel control in curbing the pandemic [10, 11]. Danon et al. modified the existing national-scale metapopulation model to establish a spatial model for the COVID-19 pandemic spread in England and Wales to predict the early and peaking times [12]. Wu et al. estimated the size of the COVID-19 cases in Wuhan and predicted the public health risks of the pandemic in China [4], and Leung et al. used a susceptible-infectious-recovered model to present the potential effects of loosening restrictions after the first-wave transmission and project potential second-wave infection [13]. These useful studies have substantially improved our understanding of the COVID-19 spread, but most of them are based on mathematical models to quantify the number of confirmed cases. It is challenging to explore the spatiotemporal changes in COVID-19 and analyze their potential influencing factors. It is of great scientific and practical significance to resolve the above questions by conducting spatiotemporal analysis of the pandemic in China through spatial analysis methods and by revealing the spatial dynamics of the pandemic regarding its occurrence, development to shrinking.

The purpose of this study is to probe the spatiotemporal spread pattern of COVID-19 in China and analyze the relationships between the number of cases and its potential influencing factors, providing brief results and valuable insights for understanding COVID-19's spatial transmission and controlling the global COVID-19 pandemic spread.

## Data and methods

### Data collection

The number of new and total COVID-19 cases and their locations are the key data to delineate the spatial spread of the pandemic. We collected the daily number of new and total COVID-19 cases, for each prefecture-level and county-level city (Chinese Mainland, Hong Kong and Macao special administrative regions and Taiwan province), reported by the National Health Commission of China and the Health Commissions of local governments from 17 January to 20 March, 2020. All data is publicly accessible at the website of the National Health Commission of China (en.nhc.gov.cn). We reclassified the data by week to better capture the critical changes in the pandemic spread while reducing the data redundancy.

To briefly illustrate the impacts of determinants on the pandemic spread, we considered the sociodemographic and human disturbance aspects. Among these, the sociodemographic impacts were represented using the total persons per pixel (PopDensi measured in remote sensing images) provided by WorldPop (www.worldpop.org), and the population movements (PopInflow) between Wuhan and other Chinese cities (qianxi.baidu.com/2020). The human disturbances were measured using the proximity to provincial cities (Dis2CaptCity),

prefecture cities (Dis2PrefCity), and intercity traffic stations (Dis2TrafStation). The proximity factors were produced using the Euclidean distance based on vector maps collected from OpenStreetMap (www.openstreetmap.org) and Baidu (qianxi.baidu.com). The potential factors were all resampled to a resolution of 500 m. Although higher resolutions could offer more details, they would also result in negative effects such as data redundancy and lower computational efficiency. The base map of China was provided by the Resource and Environment Data Cloud Platform (www.resdc.cn). All the spatial data were projected using the Albers equal-area conic projection.

## Analysis methods

To better present the COVID-19 pattern, we classified the case number into 10 levels: (0, 8], (8,16], (16, 32], (32,64], (64,128], (128,256], (256,512], (512,1024], (1024,2048], and (2048,$\infty$]. To clarify the spatial trend of the COVID-19 transmission, we applied the Anselin Local Moran's I statistic to delineate the spatial distribution of the weekly new and total cases. This statistic divides the study area into five categories of regions: 1) high-high cluster (severely infected area, cities with high cases surrounded by cities with high cases), 2) high-low outlier (spatial outlier, cities with high cases surrounded by cities with low cases), 3) low-high outlier (spatial outlier, cities with low cases surrounded by cities with high cases), 4) low-low cluster (secure area, clustered areas with similar low cases), and 5) not statistically significant area (the number of cases in cities was not statistically clustered). The clusters were defined at a significance level of 0.05. We applied the Huhuanyong Line to describe the COVID-19 patterns on the east and west sides of China, and described the impact of distance to Wuhan on the spatial pattern of COVID-19 using concentric buffer rings with Wuhan as the center.

Using a generalized additive model (GAM), we linked each potential factor with the COVID-19 cases to quantify its effect on the pandemic. Because linear regression cannot address the COVID-19 changes across space, we used the GAM's flexible smoothing method to build the complex relationships between the COVID-19 spread and its influencing factors. GAM is a nonparametric extension of the generalized linear model, which makes the model flexible to deal with the nonlinear relationship between response and multiple explanatory variables by using an unspecified smoothing function [14, 15]. The rank-order of factors plays a key role in GAM, where a prior factor is more statistically significant and indicates a stronger explanatory ability [16]. After defining the sort-order of factors, the GAM can be written as:

$$L(cases) \sim \beta_0 + s_1(\text{PopDensi}) + s_2(\text{Dis2CaptCity})$$
$$+ s_3(\text{Dis2TrafStation}) + s_4(\text{PopInflow}) + s_5(\text{Dis2PrefCity}) + \delta \tag{1}$$

where $L(cases)$ is a link function that represents the effects of influencing factors on the COVID-19 cases, $\beta_0$ donates a constant, $s_i()$ is a smoothing function that describes the relationships between $L(cases)$ and the $i^{th}$ factor, and $\delta$ is the model residual.

## Results

### Spatiotemporal distribution of cities with COVID-19 cases

**Distribution of cities confirming cases.** Fig 1 shows that the COVID-19 cases in China, which initially occurred in Wuhan and spread to all parts of China, were primarily distributed in cities on the east side of the Huhuanyong Line and only a few on the west side. The cases peaked on 7 February and were effectively curbed after 21 February. Fig 1A shows the distribution of cities where new COVID-19 cases have been reported per week. Before 17 January, the COVID-19 cases were confirmed in Wuhan only. Although the government imposed a

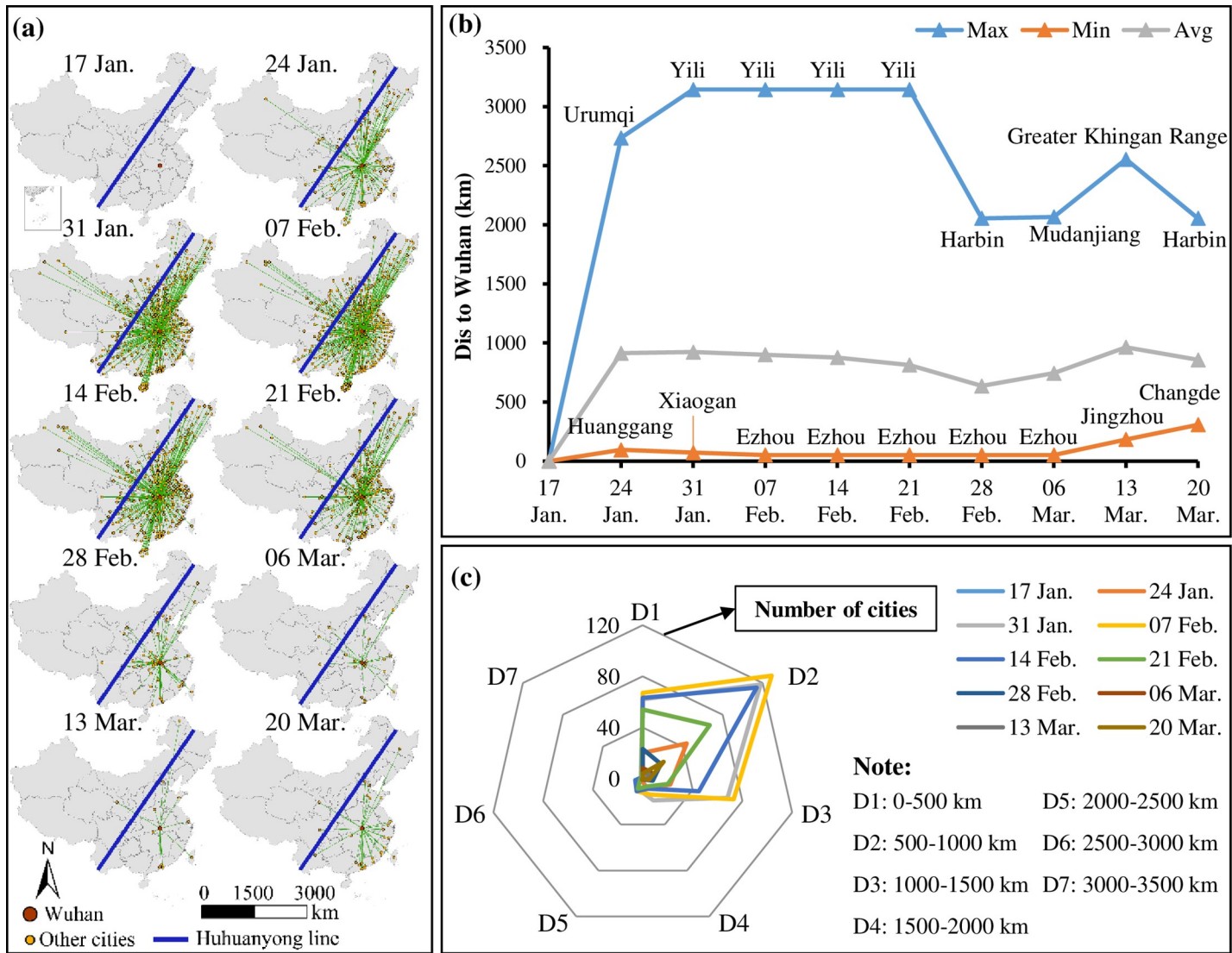

**Fig 1. Spatiotemporal pattern of cities where new COVID-19 cases were reported each week, from 17 January 2020 to 20 March 2020.** (a) Location of the cities; (b) the nearest, farthest and average distance from Wuhan to the cities with new cases; and (c) the number of cities with new COVID-19 cases at different distances to Wuhan.

lockdown on Wuhan at 10:00 am on 23 January, the COVID-19 pandemic quickly spread to surrounding cities and other areas in China during the Chinese New Year transport season, and the number of new cases increased sharply. Fig 1B shows that, as early as the week between 17–24 January, the pandemic spread to nearby cities Ezhou and Huanggang, and to other cities adjacent to Wuhan; since 24 January, the COVID-19 cases were even found in Urumqi and Ili Kazakh Autonomous Prefecture cities that are more than 2,500 km from Wuhan. In the week between 14–20 March, the COVID-19 pandemic in Hubei Province was substantially under control, and Changde of Hunan Province (300 km away from Wuhan) is the nearest city that reported new cases. Fig 1C shows that the cases are mainly distributed in regions that are less than 1,000 km from Wuhan. From the end of February, the growth rate of new cases gradually slowed down, which indicates that China effectively controlled the spread of the COVID-19 within two months.

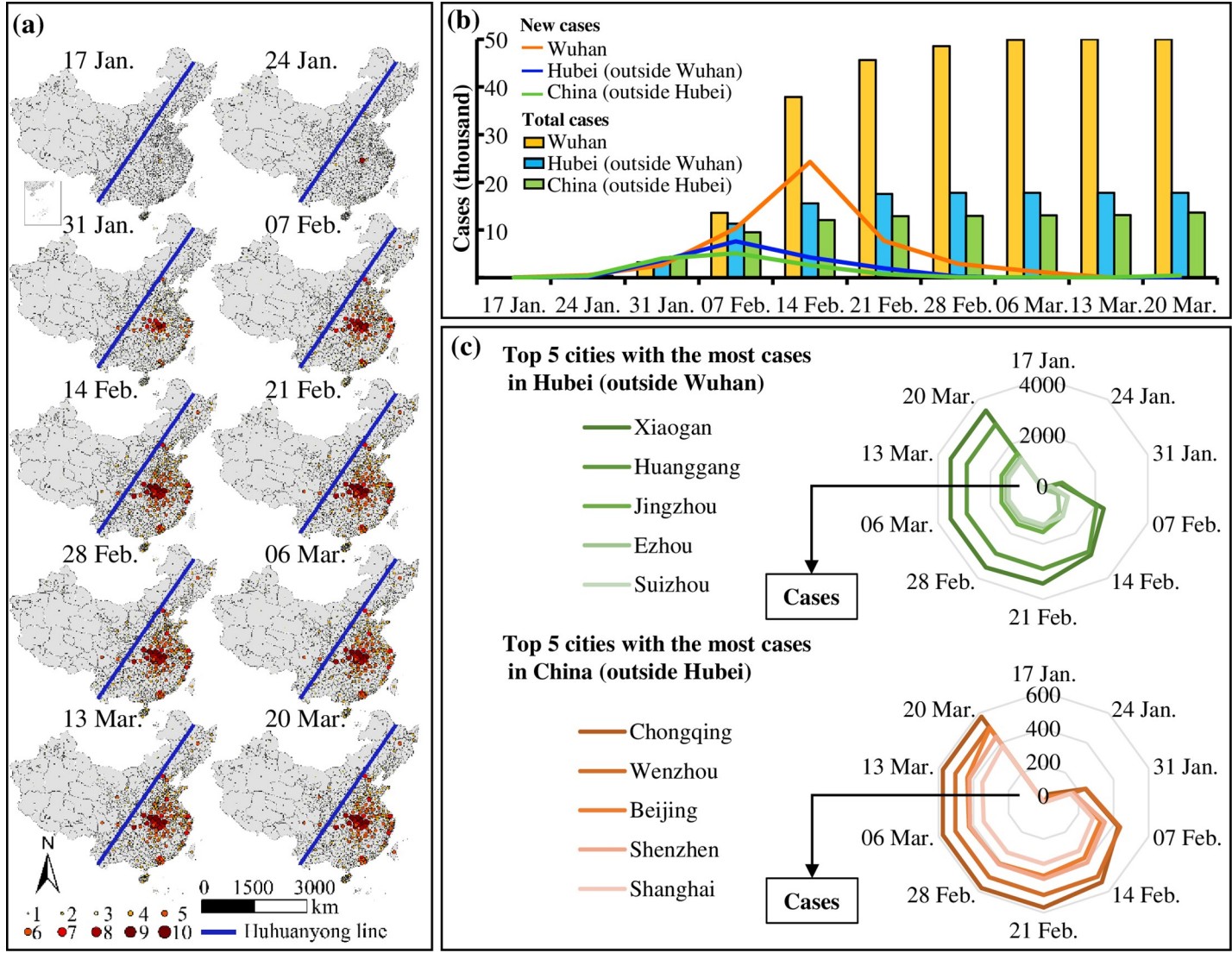

**Fig 2. Spatiotemporal pattern of the total COVID-19 cases each week, from 17 January 2020 to 20 March 2020.** (a) Levels of the number of the total cases in different cities; (b) the new and total cases in Wuhan, Hubei (outside Wuhan) and China (outside Hubei); and (c) the top 5 cities with the most cases in Hubei (outside Wuhan) and China (outside Hubei).

Fig 2 shows the spread trend of COVID-19 from Wuhan to its surroundings as well as a few metropolises with a large population of over 10 million, and that the total cases gradually stabilized at 84,000 after rapid increases in middle February. Fig 2A illustrates that most cities with high cases are located on the east side of the Huhuanyong Line while only a few cities reported cases are located on the west side. Specifically, the pandemic is serious in Wuhan and surrounding cities, and considerable cases also occurred in Guangzhou, Shenzhen, Shanghai, and Beijing, as well as cities along the coast of Zhejiang Province. Fig 2B compares the new and total cases in Wuhan, Hubei (outside Wuhan) and China (outside Hubei), where the number of new cases in Wuhan peaked in the week between 8–14 February and those in Hubei and China peaked in the week between 1–7 February. This indicates that the inflection point for Wuhan was on 14 February, one week later than those of Hubei and China. Two weeks after the first Fangcang shelter hospitals started to accept patients on 5 February, the number of

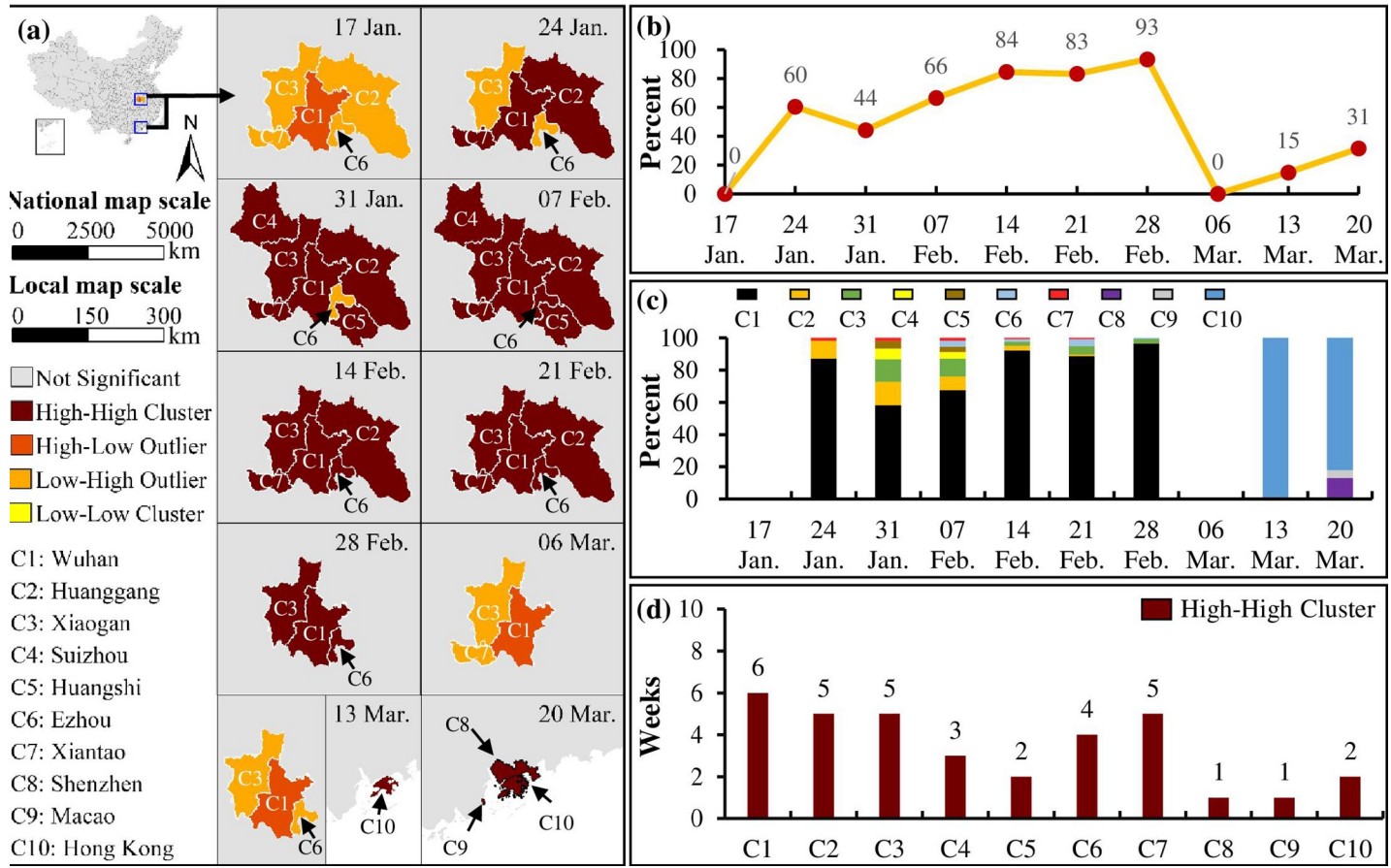

**Fig 3. Spatial clusters of the new COVID-19 cases each week from 17 January 2020 to 20 March 2020.** (a) Spatiotemporal change in the clusters of new cases; (b) the percent of new cases in the high-high clusters compared to the total new cases in China; (c) the percent of the new cases in each city in each high-high cluster; and (d) the number of weeks each city included in the high-high clusters.

new cases in Wuhan declined sharply. Fig 2C presents the top 5 cities with serious COVID-19 in Hubei (outside Wuhan), including Xiaogan, Huanggang, Jingzhou, Ezhou and Suizhou in descending order, and the top-5 cities in China (outside Hubei), including Chongqing, Wenzhou, Beijing, Shenzhen and Shanghai in descending order. These results suggest that the changes in the COVID-19 cases and the pandemic spread were significantly affected by population movements between Wuhan and the destinations.

**Spatial clustering of total cases.** Clustering analysis of the new cases shows that there was a high-high cluster (areas with serious pandemic situation in COVID-19) around Wuhan before 6 March, which moved to the Grand Bay Area (Hong Kong, Shenzhen and Macau) after 6 March (Fig 3A). The high-high cluster occurred in Wuhan, Huanggang and Xiantao on 24 January, and it expanded to include Xiaogan, Suizhou and Huangshi one week later. Then, Ezhou was jointed into the high-high cluster on 7 February when the cluster peaked in its acreage. With the alleviation of the shortage of medical resources, from 14 February to 28 February, the high-high cluster gradually shrank and it was still in Hubei. During the week between 28 February and 6 March, there was no statistically significant cluster. The high-high cluster finally occurred in the Grand Bay Area from 7 March to 20 March, including Hong Kong, Macau and Shenzhen into the cluster sequentially. Fig 3B shows that the new cases in Hubei (outside Wuhan) accounted for at least 44% (peaked with 93% on 28 February) of the total

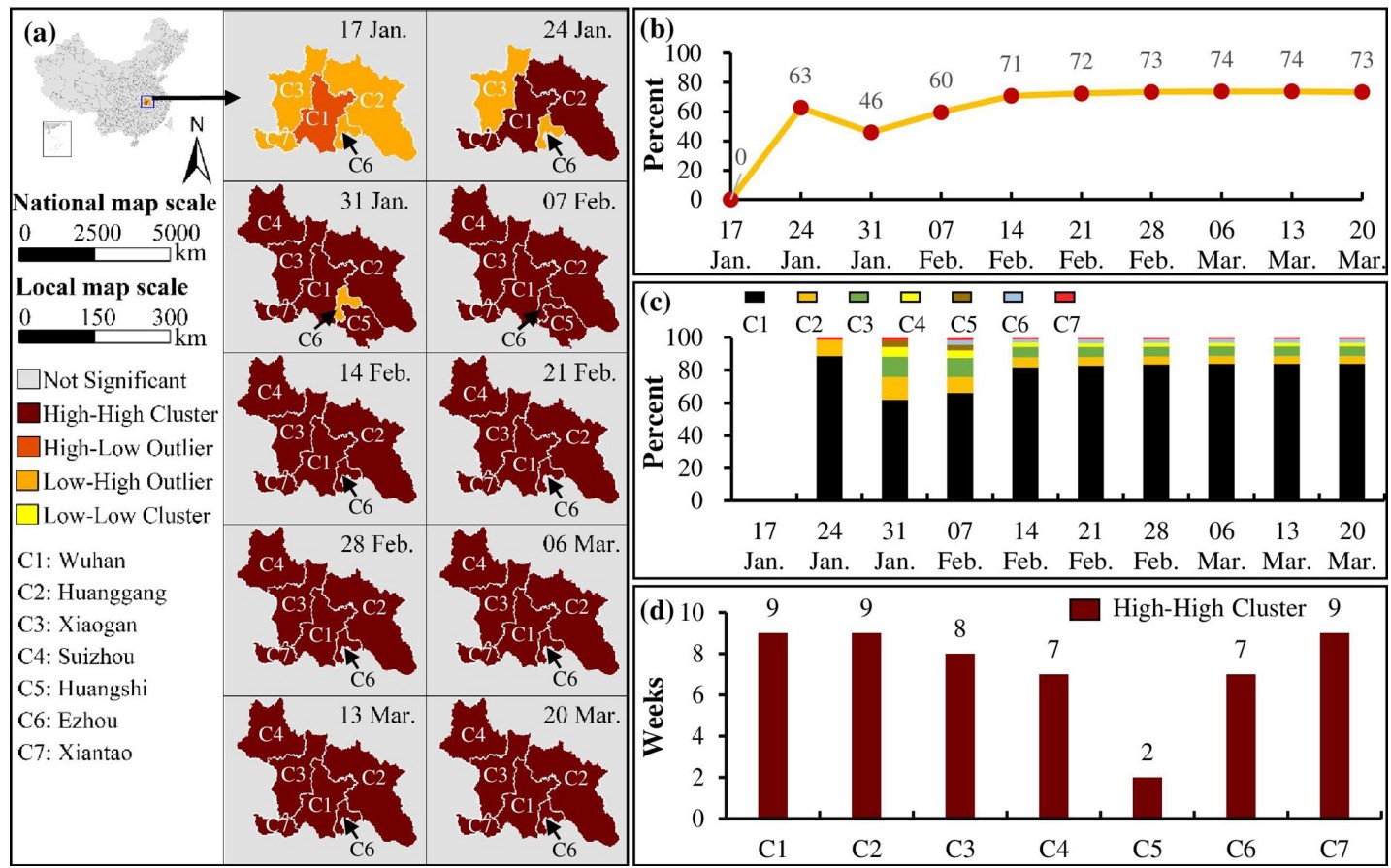

**Fig 4. Spatial clusters of the total COVID-19 cases each week from 17 January 2020 to 20 March 2020.** (a) Spatiotemporal change in the clusters of the total cases; (b) the percent of the total cases in the high-high clusters compared to the total cases in China; (c) the percent of the total cases in each city in each high-high cluster; and (d) the number of weeks each city included in the high-high clusters.

new cases in China, while the new cases in the Grand Bay Area only account for fewer than 31%. Fig 3C shows that Wuhan was ascribed to most cases in the first cluster while Hong Kong was ascribed to most cases in the second cluster. This indicates that the COVID-19 outbreak was highly concentrated in Wuhan before March and then in Hong Kong. Fig 3D illustrates that Wuhan lasted in the most serious area for 6 weeks, and Huanggang, Xiaogan and Xiantao lasted for 5 weeks, Ezhou lasted for 4 weeks, and the other cities lasted fewer than 3 weeks. These results suggest that the five cities, including Wuhan, Huanggang, Xiaogan, Xiantao and Ezhou, were the most serious areas being affected by COVID-19.

Fig 4A demonstrates that the cities with a high incidence of COVID-19 cases were clustered in Hubei, including seven cities: Wuhan, Huanggang, Xiaogan, Suizhou, Huangshi, Ezhou and Xiantao. Fig 4B indicates that COVID-19 was first reported in Wuhan, and the total cases in the high-high clusters account for more than 60% of the country's total cases on 24 January and 7 February, half in/on 31 January, and more than 70% from 14 February. When the SARS-CoV-2 virus spread in Wuhan, the citizens had a weak awareness of self-protection and participated in a few large-scale social events, such as a large neighborhood party held in the Baibuting community on 18 January. This has led to the community-level cluster infections and outbreaks of COVID-19. As a result, Wuhan's total cases account for more than 60% of the total cases in the high-high clusters from 24 January. Fig 4D shows that since 24 January,

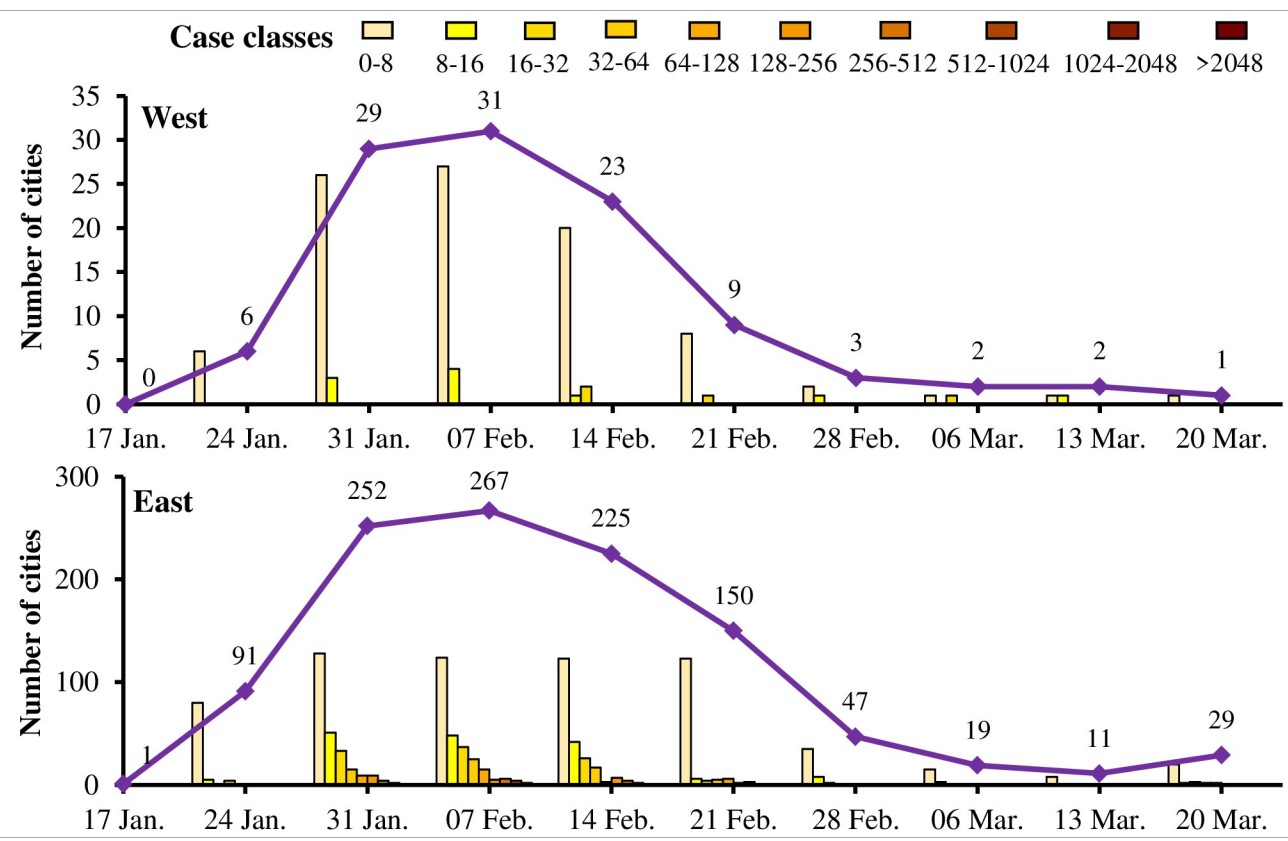

**Fig 5. Cities categorized by the levels of the new COVID-19 cases on both sides of the Huhuanyong Line.**

all cities lasted for more than 7 weeks in the high-high clusters except Huanggang. Different from the clusters of new cases, Hubei had been the most serious COVID-19 area, which contributed to the highest number of the total COVID-19 cases since the outbreak.

## Spatial pattern characterized by the Huhuanyong Line

Fig 5 indicates that the number of cities with new COVID-19 cases is much greater on the east side than on the west side of the Huhuanyong Line. During the study period, the total number of cities on the west side with new cases was less than 32, while many cities on the east side reporting more than 64 new cases per week and more than 200 cities reported new cases during 31 January and 21 February. During 7 March and 13 March, the fewest cities on the east side experienced new local cases, but there were more new cases after 13 March because of the overseas transmission cases. Since then, China has begun to guard against the overseas transmission cases to avoid a soar in potential COVID-19 infections.

Fig 6 shows that, for both sides, cities occurring COVID-19 have increased gradually from 17 January to 7 February when the number of cities peaked by 41 on the west and 286 on the east. The total number of cases in all western cities is less than 64, and most of them have no more than 8 cases. After 7 February, the pandemic situation in western cities was basically under control basically, while the situation in eastern cities continued to deteriorate, with the cities reporting much more COVID-19 cases. A few cities on the east side reported more than 500 cases. The pandemic peak of China's current pandemic has passed, but the alarm of the second spread wave in local areas (e.g. frontier cities and international metropolis) does not stop.

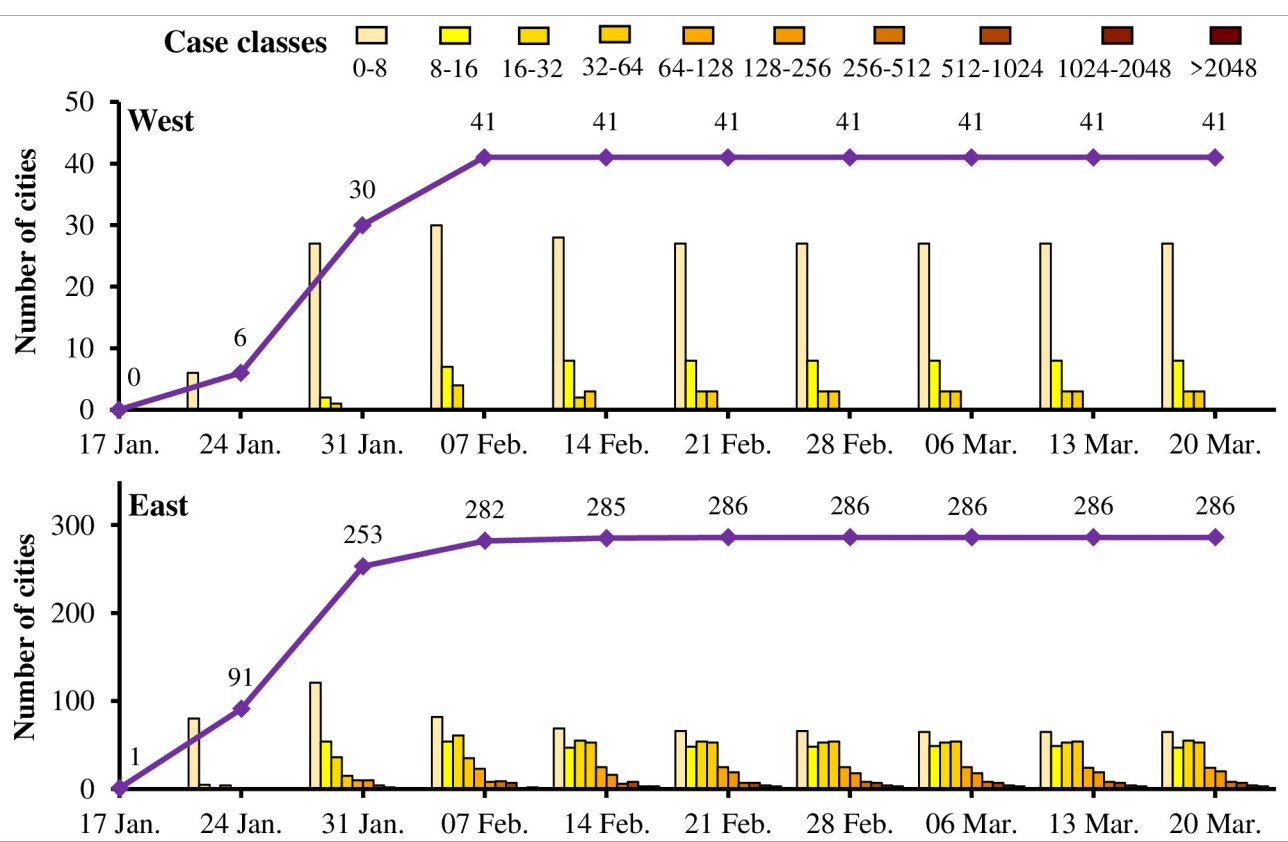

**Fig 6. Cities categorized by the levels of the total COVID-19 cases on both sides of the Huhuanyong Line.**

### Impact of distance to Wuhan

Fig 7 shows that the distance to Wuhan substantially affects the occurrence of the COVID-19 cases as revealed by the 200 km radius-based buffer rings. There are three distinct periods of the COVID-19 spread: 1) Period 1 from 17 January to 24 January (Fig 7A), 2) Period 2 from

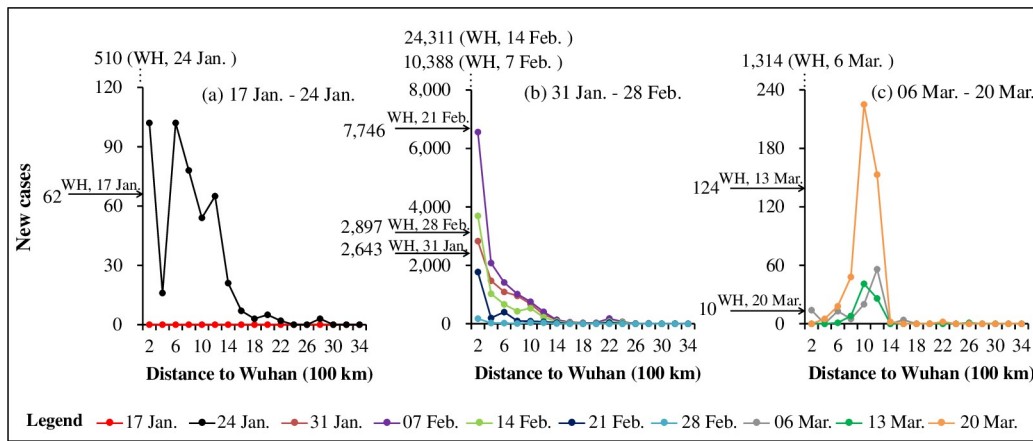

**Fig 7. The distribution of weekly new COVID-19 cases in cities with distance to Wuhan.** (a) Period 1 from 17 January to 24 January; (b) Period 2 from 31 January to 28 February; and (c) Period 3 from 6 March to 20 March.

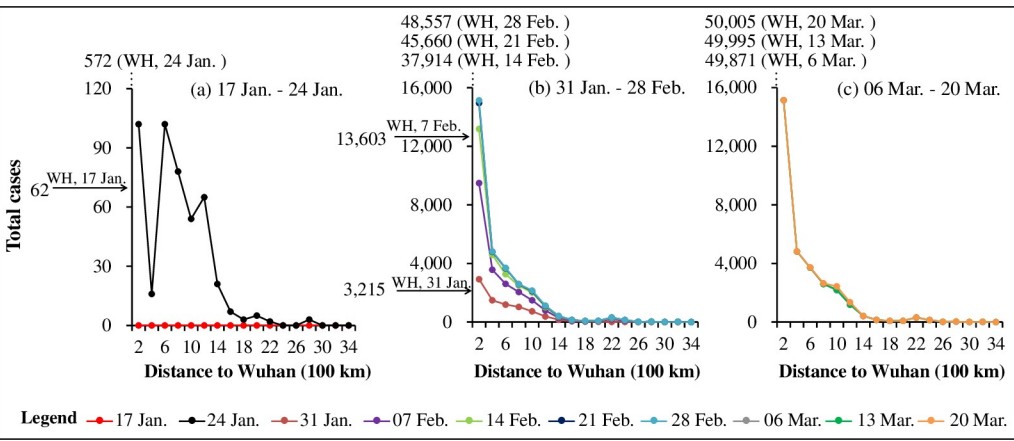

**Fig 8. The distribution of the total COVID-19 cases with distance to Wuhan.** (a) Period 1 from 17 January to 24 January; (b) Period 2 from 31 January to 28 February; and (c) Period 3 from 6 March to 20 March.

25 January to 28 February (Fig 7B), and 3) Period 3 from 29 February to 24 March (Fig 7C). In the period 1, cities with distances of 200~400 km to Wuhan reported few new cases, and those with distances of 600~1200 km to Wuhan reported more cases (Fig 7A), which may be due to the poor virus detection ability at the beginning of the pandemic. In the period 2, the new cases in Wuhan increased dramatically and a closer region had more new cases each week (Fig 7B). The curve also shows a small bulge at 2,200 km, which is attributed to the new cases of the northeast city Harbin around February 7. COVID-19 was more severe in the small city of Jining than in megacities such as Hangzhou and Zhengzhou within a distance to Wuhan (600~800 km). In the week of 8–14 February, new cases in most cities decreased compared with the last week, but the new cases in Wuhan increased sharply, which can be attributed to the new scheme to diagnose the pandemic and more than 10,000 were reported on 12 February. Then the new cases in Hubei began to decrease, with the assistance of medical staff and resources across the country. In the period 3, except Wuhan, cities with distances of 800~1400 km from Wuhan reported the most of the new cases (Fig 7C). Although nearby cities such as Huanggang and Xiaogan had high total cases, they had no increase later (Fig 7C). New cases that surged in Shenzhen and Beijing were largely due to the overseas imported cases in the week of 16–20 March. Overall, for local cases in China, the distance to Wuhan substantially reflects the regional population flow and thus the spatial diffusion of the infected COVID-19 cases across cities.

Fig 8 shows that except for cities (e.g. Beijing, Shanghai, Wenzhou and Guangzhou) with large population flow with Wuhan, within a distance of 1,600 km, the closer to Wuhan the greater the number of total cases. The total cases have weak relations with the distance to Wuhan when it is further than 1,600 km. Fig 8A shows that in the early stage of the outbreak, the distribution of total cases was similar to that of the new cases (Fig 7A). Fig 8B and 8C show that the total cases have been significantly affected by the distance to Wuhan since 25 January, Chinese New Year. There are thousands of cases in Xiaogan, Xiangyang and Huanggang, nearby cities of Wuhan; while the Jiamusi, Altay and Rikaze cities that are far away from Wuhan had very few cases. The curves of the last two weeks show minor changes, indicating that the COVID-19 cases in China have already peaked.

## Impact of potential influencing factors

We further analyzed the impact of potential influencing factors on the COVID-19 spread using GAM. The degree of freedom (DOF) of the GAM's smoothing function has a substantial

**Table 1. The best degrees of freedom of the potential influencing factors using AIC.**

| Factors | DOF | Implication |
| --- | --- | --- |
| PopDensi | 8.65 | A strong relationship between the total COVID-19 cases and the factor. |
| Dis2CaptCity | 7.83 | A strong relationship between the total COVID-19 cases and the factor. |
| Dis2TrafStation | 1 | A weak relationship between the total COVID-19 cases and the factor. |
| PopInflow | 1 | A weak relationship between the total COVID-19 cases and the factor. |
| Dis2PrefCity | 1 | A weak relationship between the total COVID-19 cases and the factor. |

influence on the model's estimation and stability. We applied the Akaike Information Crite-rion (AIC) to define the best DOF to address the potential influencing factors (Table 1).

The loess plots (95% confidence level) show that the relationships between the total cases and the five potential factors (Fig 9). The PopDensi and Dis2CaptCity have higher DOF values, which significantly associated with the total number of cases. The Dis2PrefCity, Dis2TrafSta-tion and PopInflow factors have a DOF of 1, which indicates their weak correlations with the total cases. The provincial capitals outside Hupei have more cases than other prefecture-level cities, where the capitals have higher population density and more frequent population move-ment than the prefecture-level cities, relating to higher cases in the capitals as revealed by the PopDensi and Dis2CaptCity factors. In prefecture-level cities, some counties have serious pan-demics, while others may have no cases; thus, the total number of cases is much weakly related

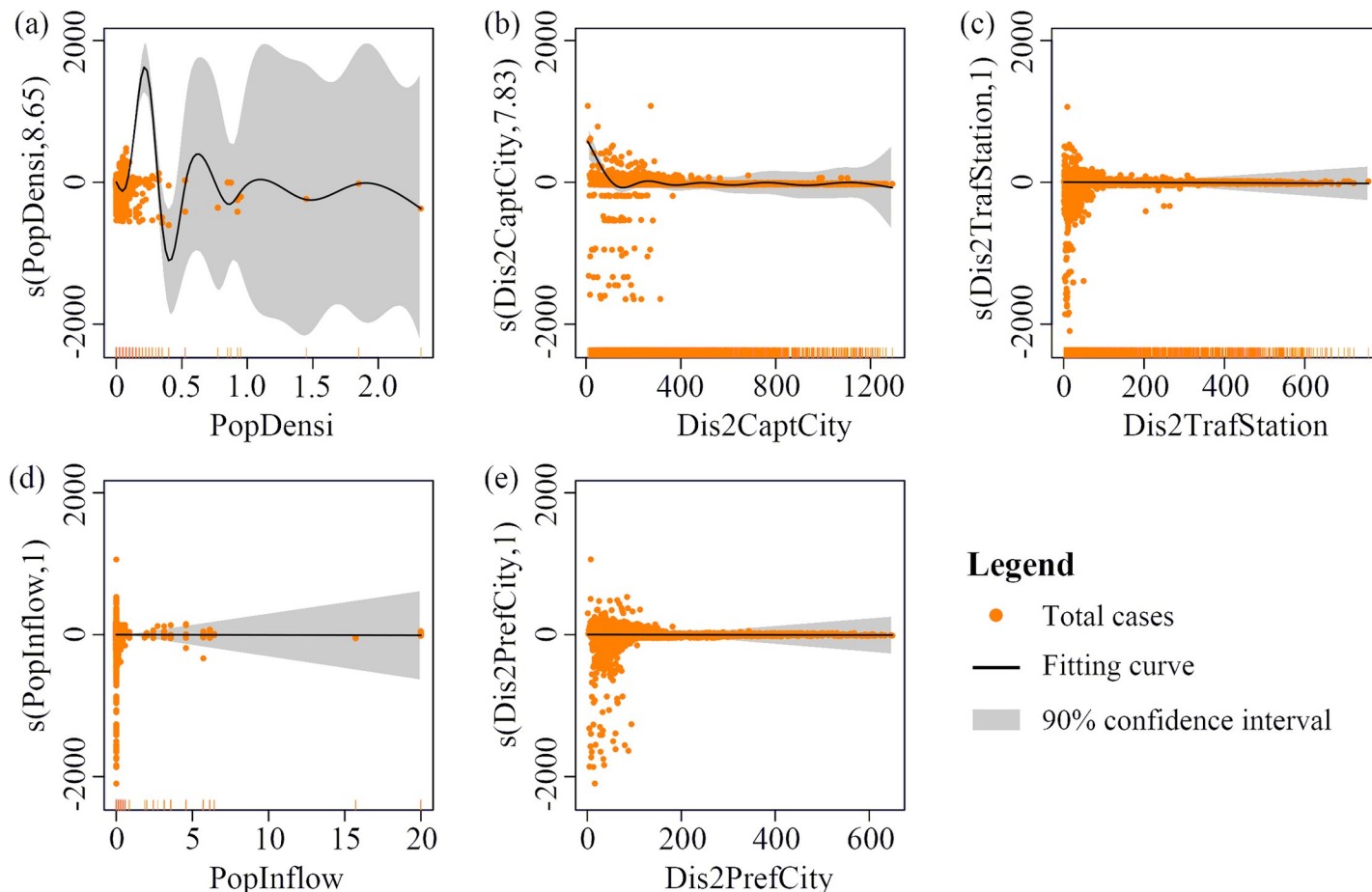

**Fig 9. Relationships between the number of COVID-19 cases and their potential impact factors with 95% confidence level.**

to Dis2PrefCity (the distance to the prefecture-level city centers). Since 23 January, the population outflow from Wuhan has been extremely minor because of the lockdown and the strict quarantine measures, resulting in a weak relationship between the total COVID-19 cases and PopInflow and Dis2TrafStation. However, the outflow of the population before lockdown and the long incubation period of the disease caused the rapid spread of the pandemic later.

## Discussion

The outbreak of COVID-19 is a great challenge to people's health, medical systems, economic development and social stability. It is valuable to conduct spatial analysis of COVID-19 in China in prefecture-level cities. This study used data from the weekly new and total cases to delineate the spatiotemporal pattern and dispersal trend of COVID-19 during 17 January (the first day Wuhan officially reported the COVID-19 cases) and 20 March (no new domestic COVID-19 cases).

Our analysis shows that the COVID-19 pandemic spread in China is closely associated with geographic factors and human activities. In China, the pandemic first broke out in Wuhan, and subsequently Hubei and its neighboring Henan, Hunan, Anhui, Jiangxi, and Chongqing became the core affected areas. Anselin Local Moran's I statistic results show the strong spatial correlation. The number of COVID-19 cases in densely populated areas may increase, leading to the spread of COVID-19 to other areas [17, 18]. In the early days of the pandemic, the cities with a high number of cases are provincial capitals (e.g. Changsha of Hunan) and transportation hubs (e.g. Bengbu of Anhui). The pandemic then rapidly spread in these cities initially caused by the imported transmission from Wuhan and then by local transmission. For cities such as Jining (in Shandong) and Kunming (in Yunnan) that are far away from Hubei, COVID-19 has spread rapidly in local transmission because the public and personal preventive measures were not sufficient at first. Early seed cases in other parts of China originated in Wuhan and many cases after April, 2020 were from overseas. However, the global outbreak of COVID-19 could not be originated in Wuhan because the Patient Zero do not always come from the location where the first cluster of cases occurs. The birthplace of the COVID-19 pandemic is still being explored by biologists around the world [1–3].

According to the WHO, the infection is mainly transmitted through contact with respiratory droplets that are generally caused by short social distance [19]; macroscopically, COVID-19 in China is greatly affected by population movements between Wuhan and other cities and the subsequent local transmission. Thus, personal preventive measures such as wearing a mask and maintaining a long social distance are very important to protect each individual and others from being infected. A flood of people leaving Wuhan during the Chinese New Year could spread the virus across the country before all entrances and exits were closed on 23 January. Four metropolises (e.g., Beijing, Shanghai, Shenzhen, and Guangzhou) that are more than 700 km from Wuhan experienced serious pandemic, where many passengers went in and out of Wuhan. As one of China's major trading cities, Wenzhou has many businessmen working in Wuhan who returned home before Chinese New Year (www.wenzhou.gov.cn), ascribing to the rapid spread of COVID-19 in this city. Harbin and Sanya, which are far away (more than 1,400 km) from Wuhan. For example, they also unexpectedly experienced a serious COVID-19 pandemic (news.china.com.cn), which may be attributed to the tourists coming from Wuhan, and some of them have carried the SARS-CoV-2 virus [20]. Population migration is closely related to the deterioration of pandemic, and it is a harbinger of the future situation of epidemics [21, 22].

For the analysis of macro impacts, we considered population density, population inflow from Wuhan, and distances to provincial cities, prefecture-level cities and traffic stations. The

findings revealed that population density and distance to provincial capitals are the most influential factors of the COVID -19 spread while distance to transportation facilities and prefecture-level cities, population flows only weakly affect the COVID-19 spread. According to the authoritative records, many cases were initially found in stations, docks and airports, but these may not be reflected on the map and only reported in the statistics of the new and total cases (www.shanghai.gov.cn; www.hainan.gov.cn). After the lockdown of Wuhan, population outflow was minor; thus, the data we acquired from Baidu migration (in percentage) may be insufficient to conduct a systematic analysis. Because COVID-19 has an incubation period of even more than 40 days, these people outflow from Wuhan could cause community-level infections in other cities. Timely travel restrictions have weakened the spread of the pandemic [22, 23]. Population social gathering together with the infection source from exposure in Wuhan has occurred in many cities such as Tianjin, Beijing and Chengdu, which was an essential cause leading to the rapid deterioration of the COVID-19. The pandemic distribution in China has significant spatial heterogeneity, which coincides with results of Fronterre, Read [24] in England.

It was acknowledged that nonpharmaceutical public health interventions can effectively curb the spread of COVID-19 [25]. From 23 January, Wuhan began to build two makeshift hospitals, Huoshenshan (Fire God Mountain) and Leishenshan (Thunder God Mountain) in 10 days to treat the severe COVID-19 patients by replicating Beijing's SARS treatment model in 2003. However, the shortage of medical resources and weak protective measures led to an explosive increase in the number of cases in the early stages of the pandemic (Figs 1 and 2). To treat the mild COVID-19 patients, the first three Fangcang shelter hospitals were built in 3 days since its construction started on 2 February. The number of new cases in Wuhan reached its maximum in the next two weeks and then began to decrease (Fig 2). From 5 February to 10 March, a total of 16 Fangcang shelter hospitals opened in Wuhan to provide isolation, detection, treatment and shelter for patients with mild symptoms of COVID-19, curing more than 12,000 patients [26]. More than 42,000 medical personnel from all over the country gathered to fight against COVID-19 with local medics in Wuhan, and China has established a "pairing up support for Hubei" relationship, which requested provinces and municipalities to support 16 cities in Hubei. These greatly eased the pressure on the Wuhan medical system and effectively prevented the further spread of the pandemic. All levels of financial departments have allocated 121.8 billion Chinese Yuan for pandemic prevention and control, of which the Central Ministry of Finance spent 25.75 billion Chinese Yuan as of 21 March [27]. All these medical staff and medical resources are the main reasons for the rapid control of COVID-19 in China.

In addition, community-based countermeasures are strictly carried out in all cities, which include suspending mass gathering, and using online health QR codes to enter and exit communities and public places. The government guides the residents to enhance self-protection, maintain on-and-off social distancing, and comply with the prevention and control regulations according to laws. Some unprecedented moves have also been introduced, such as extending the Spring Festival holiday and postponing the spring semester of schools and universities [28]. These social measures also ensured that the pandemic in China was basically under control, and there was no significant rebound. At present, the COVID-19 pandemic in China has been controlled substantially, but yet to be eliminated completely. Most recently, many overseas imported cases were confirmed in Harbin of northeast China and Guangzhou of southern China. This may lead to the second pandemic wave and affect the spread pattern of the pandemic in China, but the recent medical experiences and the increased awareness of the public will greatly reduce the spread speed and scope of the pandemic. China's pandemic

prevention and control measures can provide a powerful reference for other parts of the world to curb the spread of the pandemic as soon as possible.

## Conclusions

This study analyzes the spatiotemporal spread pattern of COVID-19 in China, and the impacts of potential influencing factors on the pandemic spread. Our contributions and conclusions are as follows: 1) the COVID-19 cases were mainly distributed in the east of the Huhuanyong Line, where the epicenters account for a high proportion of the country's total; 2) restricted by medical resources and the ability to detect the SARS-CoV-2 virus, the inflection point of COVID-19 in Wuhan was on 14 February, one week later than Hubei (outside Wuhan) and China (outside Hubei); and 3) population density and distance to provincial cities were significantly associated with the total number of cases as revealed by the GAM analysis. On-and-off social distancing is a powerful and cost-effective approach until targeted vaccines and specific drugs are available. All results and findings provide valuable insights into the transmission evolution and curbing the spread of COVID-19.

## Author Contributions

**Conceptualization:** Yongjiu Feng.

**Data curation:** Sicong Liu, Yanmin Jin.

**Formal analysis:** Rong Wang, Shuting Zhai, Chen Gao, Zhenkun Lei, Shurui Chen.

**Investigation:** Chen Gao, Huan Xie, Kuifeng Luan, Jinfu Xu, Hua Jiang.

**Methodology:** Chen Gao, Zhenkun Lei, Shurui Chen, Xiongfeng Yan, Zhonghua Hong, Jinfu Xu.

**Resources:** Shuting Zhai, Yilun Zhou, Jiafeng Wang, Xiongfeng Yan, Huan Xie, Shijie Liu, Xiong Xv, Chao Wei, Changjiang Xiao, Yiyou Guo.

**Software:** Rong Wang, Chao Wang.

**Supervision:** Yongjiu Feng, Xiaohua Tong.

**Validation:** Peng Chen.

**Visualization:** Shijie Liu.

**Writing – original draft:** Qingmei Li.

**Writing – review & editing:** Yongjiu Feng, Qingmei Li, Xiaohua Tong.

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
