## [Decision Letter · Decision Letter 0]

25 Sep 2020

PONE-D-20-27005

Spatiotemporal spread pattern of the COVID-19 cases in China

PLOS ONE

Dear Dr. Feng,

Thank you for submitting your manuscript to PLOS ONE. After careful consideration, we feel that it has merit but does not fully meet PLOS ONE’s publication criteria as it currently stands. Therefore, we invite you to submit a revised version of the manuscript that addresses the points raised during the review process.

We look forward to receiving your revised manuscript.

Kind regards,

Bing Xue, Ph.D.

Academic Editor

PLOS ONE

Journal Requirements:

2. In your ethics statement in the Methods section and in the online submission form, please provide additional information about the data used in your retrospective study. Specifically, please ensure that you have discussed whether all data were fully anonymized before you accessed them and whether the data is all publicly accessible.

"The funders had no role in study design, data collection and analysis, decision to publish, or preparation of the manuscript.".

5.  We note that Figures in your submission contain map images which may be copyrighted. All PLOS content is published under the Creative Commons Attribution License (CC BY 4.0), which means that the manuscript, images, and Supporting Information files will be freely available online, and any third party is permitted to access, download, copy, distribute, and use these materials in any way, even commercially, with proper attribution. For these reasons, we cannot publish previously copyrighted maps or satellite images created using proprietary data, such as Google software (Google Maps, Street View, and Earth). For more information, see our copyright guidelines: http://journals.plos.org/plosone/s/licenses-and-copyright.

1.     You may seek permission from the original copyright holder of Figures to publish the content specifically under the CC BY 4.0 license.  

Reviewers' comments:

Reviewer's Responses to Questions

**Comments to the Author**

1. Is the manuscript technically sound, and do the data support the conclusions?

Reviewer #1: Yes

Reviewer #2: Partly

2. Has the statistical analysis been performed appropriately and rigorously? 

Reviewer #1: Yes

Reviewer #2: No

3. Have the authors made all data underlying the findings in their manuscript fully available?

Reviewer #1: Yes

Reviewer #2: No

4. Is the manuscript presented in an intelligible fashion and written in standard English?

Reviewer #1: Yes

Reviewer #2: Yes

5. Review Comments to the Author

Reviewer #1: The authors analyzed the spatiotemporal spread pattern of covid-19 cases by means of spatial clustering, and expounded the influence of spatial distance from Wuhan on the transmission of pandemic. The results and findings should provide valuable insights for understanding the changes in the covid-19 transmission and controlling the global covid-19 panoramic spread.

I recommend to publish this paper after addressing the following issues.

(1) Authors used a generalized additive model (GAM), linked each potential factor with the COVID-19 cases to quantify its effect on the pandemic. I suggest enrich the quantitative expression of the conclusion, not just the visual expression of figure.

(2) In the discussion area, the author described the policies and measures to block the transmission of covid-19 issued by a large number of countries. I suggest that this part should be linked with the evolution of spatiotemporal transmission pattern of covid-19 cases, rather than discuss independently.

Reviewer #2: The paper entitled “Spatiotemporal spread pattern of the COVID-19 cases in China” deal with the spatiotemporal pattern of the COVID-19 pandemic in China, reveals China’s epicenters of the pandemic through spatial clustering. The paper has some shortcomings to improve.

1. Introduction: introduction contains a listing of results from previous studies, without any logical progression of thinking. It is not possible at the end of introduction to identify any research question (i.e. current challenges, promising paths to be explored, and current study contribution).

2. There are a lot of literatures about this topic. What is the innovation of this research?

3. It lacks a real Implication session to discuss what the results can be used to, and the implications of their findings to policy makers are also not explained.

6. PLOS authors have the option to publish the peer review history of their article (what does this mean?). If published, this will include your full peer review and any attached files.

Reviewer #1: No

Reviewer #2: No

---

## [Author Response · Author response to Decision Letter 0]

1 Nov 2020

PONE-D-20-27005: Spatiotemporal spread pattern of the COVID-19 cases in China

Responses to Comments:

We would like to thank the Editor and two anonymous reviewers for the constructive comments and corrections. We have considered all comments and addressed the issues in the revision, with all changes highlighted. This report presents our responses to the comments and how we have addressed them. We now resubmit our manuscript to pursue publication in PLOS ONE.

Comments from the editors and reviewers:

-Journal Requirements

Reply: Thank you very much for your decision and constructive comments. We have modified the format of this paper and the reference citation as PLOS ONE's style templates.

2. In your ethics statement in the Methods section and in the online submission form, please provide additional information about the data used in your retrospective study. Specifically, please ensure that you have discussed whether all data were fully anonymized before you accessed them and whether the data is all publicly accessible.

Reply: Thank you for the comments. All of the data in this study are publicly accessible, and the data sources have been identified in the corresponding location in the article.

"The funders had no role in study design, data collection and analysis, decision to publish, or preparation of the manuscript.".

Reply: This has been included in the revision.

4. We note that Figures in your submission contain map images which may be copyrighted. All PLOS content is published under the Creative Commons Attribution License (CC BY 4.0), which means that the manuscript, images, and Supporting Information files will be freely available online, and any third party is permitted to access, download, copy, distribute, and use these materials in any way, even commercially, with proper attribution. For these reasons, we cannot publish previously copyrighted maps or satellite images created using proprietary data, such as Google software (Google Maps, Street View, and Earth).

Reply: Thank you for the comments. The figures we submitted contain map images that are available to the public and are described in the data collection section. Among these, the sociodemographic impacts were represented using the total persons per pixel (PopDensi measured in remote sensing images) provided by WorldPop (www.worldpop.org), and the population movements (PopInflow) between Wuhan and other Chinese cities (qianxi.baidu.com/2020). The human disturbances were measured using the proximity to provincial cities (Dis2CaptCity), prefecture cities (Dis2PrefCity), and intercity traffic stations (Dis2TrafStation). The proximity factors were produced using the Euclidean distance based on vector maps collected from OpenStreetMap (www.openstreetmap.org) and Baidu (qianxi.baidu.com). The base map of China was provided by the Resource and Environment Data Cloud Platform (www.resdc.cn). 

-Reviewer 1

The authors analyzed the spatiotemporal spread pattern of covid-19 cases by means of spatial clustering, and expounded the influence of spatial distance from Wuhan on the transmission of pandemic. The results and findings should provide valuable insights for understanding the changes in the covid-19 transmission and controlling the global covid-19 panoramic spread. I recommend to publish this paper after addressing the following issues. 

Reply: Thank you for your helpful comments. We have properly addressed all the issues and carefully revised the paper.

(1) Authors used a generalized additive model (GAM), linked each potential factor with the COVID-19 cases to quantify its effect on the pandemic. I suggest enrich the quantitative expression of the conclusion, not just the visual expression of figure.

Reply: Thanks for your comments. We used as much quantitative data as possible to describe the impact of potential factors on the number of confirmed COVID-19 cases. The loess plots (95% confidence level) show that the relationships between the total cases and the five potential factors (Fig 9). The PopDensi and Dis2CaptCity have higher DOF values, which significantly associated with the total number of cases. The Dis2PrefCity, Dis2TrafStation and PopInflow factors have a DOF of 1, which indicates their weak correlations with the total cases. The provincial capitals outside Hupei have more cases than other prefecture-level cities, where the capitals have higher population density and more frequent population movement than the prefecture-level cities, relating to higher cases in the capitals as revealed by the PopDensi and Dis2CaptCity factors. In the prefecture-level cities, some counties have serious pandemics, while others may have no cases; thus, the total number of cases is much weakly related to Dis2PrefCity (the distance to the prefecture-level city centers). Since 23 January, the population outflow from Wuhan has been extremely minor because of the lockdown and the strict quarantine measures, resulting in a weak relationship between the total COVID-19 cases and PopInflow and Dis2TrafStation. However, the outflow of the population before lockdown and the long incubation period of the disease caused the rapid spread of the pandemic later.

Fig 9. Relationships between the number of COVID-19 cases and their potential impact factors with 95% confidence level.

(2) In the discussion area, the author described the policies and measures to block the transmission of covid-19 issued by a large number of countries. I suggest that this part should be linked with the evolution of spatiotemporal transmission pattern of covid-19 cases, rather than discuss independently.

Reply: Thanks for your support and useful comments. We have fully adopted your suggestions and revised the paper as much as we can. In the discussion area, we combined the data analysis results to link the national policies and measures to curb the spread of COVID-19 with the evolution of the spatiotemporal spread of covid-19 cases during the important periods.

From 23 January, Wuhan began to build two makeshift hospitals, Huoshenshan (Fire God Mountain) and Leishenshan (Thunder God Mountain) in 10 days to treat the severe COVID-19 patients by replicating Beijing’s SARS treatment model in 2003. However, the shortage of medical resources and weak protective measures led to an explosive increase in the number of cases in the early stages of the pandemic (Figs 1 and 2). To treat the mild COVID-19 patients, the first three Fangcang shelter hospitals were built in 3 days since its construction started on 2 February. The number of new cases in Wuhan reached its maximum in the next two weeks and then began to decrease (Fig 2). From 5 February to 10 March, a total of 16 Fangcang shelter hospitals opened in Wuhan to provide isolation, detection, treatment and shelter for patients with mild symptoms of COVID-19, curing more than 12,000 patients [1]. More than 42,000 medical personnel from all over the country gathered to fight against COVID-19 with local medics in Wuhan, and China has established a "pairing up support for Hubei" relationship, which requested provinces and municipalities to support 16 cities in Hubei. These greatly eased the pressure on the Wuhan medical system and effectively prevented the further spread of the pandemic. All levels of financial departments have allocated 121.8 billion Chinese Yuan for pandemic prevention and control, of which the Central Ministry of Finance spent 25.75 billion Chinese Yuan as of 21 March [2]. All these medical staff and medical resources are the main reasons for the rapid control of COVID-19 in China.

Fig 1. Spatiotemporal pattern of cities where new COVID-19 cases were reported each week, from 17 January 2020 to 20 March 2020. (a) Location of the cities; (b) the nearest, farthest and average distance from Wuhan to the cities with new cases; and (c) the number of cities with new COVID-19 cases at different distances to Wuhan.

Fig 2. Spatiotemporal pattern of the total COVID-19 cases each week, from 17 January 2020 to 20 March 2020. (a) Levels of the number of the total cases in different cities; (b) the new and total cases in Wuhan, Hubei (outside Wuhan) and China (outside Hubei); and (c) the top 5 cities with the most cases in Hubei (outside Wuhan) and China (outside Hubei).

-Reviewer #2: 

The paper entitled “Spatiotemporal spread pattern of the COVID-19 cases in China” deal with the spatiotemporal pattern of the COVID-19 pandemic in China, reveals China’s epicenters of the pandemic through spatial clustering. The paper has some shortcomings to improve.

Reply: Thanks for your beneficial comments. We have carefully considered your suggestions and revised the paper as much as possible. The latest paper should make readers to better understand our study.

1. Introduction: introduction contains a listing of results from previous studies, without any logical progression of thinking. It is not possible at the end of introduction to identify any research question (i.e. current challenges, promising paths to be explored, and current study contribution).

Reply: Thanks for your helpful comments. We rearranged the logical relationship in the introduction, and pointed out that the current research on the epidemic is mainly a quantitative analysis based on mathematical models. 

These useful studies have substantially improved our understanding of the COVID-19 spread, but most of them are based on mathematical models to quantify the number of confirmed cases. It is challenging to explore the spatiotemporal changes in COVID-19 and analyze their potential influencing factors. It is of great scientific and practical significance to resolve the above questions by conducting spatiotemporal analysis of the pandemic in China through spatial analysis methods and by revealing the spatial dynamics of the pandemic regarding its occurrence, development to shrinking.

2. There are a lot of literatures about this topic. What is the innovation of this research?

Reply: Thank you for the comments. We believe that this study has the following innovations compared with the currently published literatures:

• From the geographical perspective, we visualized the distribution characteristics of cases in China with the help of the Huhuanyong Line, and systematically analyzed the changes in the number of confirmed cases on a weekly basis.

• This study reveals China’s epicenters of the pandemic through spatial clustering, and delineates the substantial effect of distance to Wuhan on the pandemic spread.

• We used a GAM to analyze the relationships between the potential influencing factors and the number of confirmed cases. The result shows that population density and distance to provincial cities were significantly associated with the total number of the cases, while distances to prefecture cities and intercity traffic stations, and population inflow from Wuhan after 24 January, had no strong relationships with the total number of cases.

3. It lacks a real Implication session to discuss what the results can be used to, and the implications of their findings to policy makers are also not explained.

Reply: Thanks for the constructive suggestions. In the last part of the discussion, we supplemented the use of the findings and provided decision-making suggestions and references for policy makers. 

Community-based countermeasures are strictly carried out in all cities, which include suspending mass gathering, and using online health QR codes to enter and exit communities and public places. The government guides the residents to enhance self-protection, maintain on-and-off social distancing, and comply with the prevention and control regulations according to laws. These social measures also ensured that the pandemic in China was basically under control, and there was no significant rebound, which can provide a powerful reference for other parts of the world to curb the spread of the pandemic as soon as possible.

 

Reference:

1. Patients and medical workers bid farewell before final Fangcang makeshift hospital shuts down. People's Daily Online.2020 [cited 2020 April 20]. Available from: http://en.people.cn/n3/2020/0314/c98649-9668313.html.

2. Various levels of finance have allocated 121.8 billion yuan for epidemic prevention and control. China News Service.2020 [cited 2020 April 20]. Available from: www.chinanews.com/cj/2020/03-24/9136320.shtml

---

## [Decision Letter · Decision Letter 1]

9 Dec 2020

Spatiotemporal spread pattern of the COVID-19 cases in China

PONE-D-20-27005R1

Dear Dr. Feng,

We’re pleased to inform you that your manuscript has been judged scientifically suitable for publication and will be formally accepted for publication once it meets all outstanding technical requirements.

Kind regards,

Bing Xue, Ph.D.

Academic Editor

PLOS ONE

Additional Editor Comments (optional):

Reviewers' comments:

Reviewer's Responses to Questions

**Comments to the Author**

1. If the authors have adequately addressed your comments raised in a previous round of review and you feel that this manuscript is now acceptable for publication, you may indicate that here to bypass the “Comments to the Author” section, enter your conflict of interest statement in the “Confidential to Editor” section, and submit your "Accept" recommendation.

Reviewer #2: All comments have been addressed

2. Is the manuscript technically sound, and do the data support the conclusions?

Reviewer #2: Yes

3. Has the statistical analysis been performed appropriately and rigorously? 

Reviewer #2: Yes

4. Have the authors made all data underlying the findings in their manuscript fully available?

Reviewer #2: Yes

5. Is the manuscript presented in an intelligible fashion and written in standard English?

Reviewer #2: No

6. Review Comments to the Author

Reviewer #2: The author has revised it according to the revised opinions，please improve the English language expression level

7. PLOS authors have the option to publish the peer review history of their article (what does this mean?). If published, this will include your full peer review and any attached files.

Reviewer #2: No

---

## [Editor Report · Acceptance letter]

22 Dec 2020

PONE-D-20-27005R1 

Spatiotemporal spread pattern of the COVID-19 cases in China 

Dear Dr. Feng:

I'm pleased to inform you that your manuscript has been deemed suitable for publication in PLOS ONE. Congratulations! Your manuscript is now with our production department. 

Kind regards, 

on behalf of

Professor Bing Xue 

Academic Editor

PLOS ONE